# Drug repositioning for pan-cancers of the digestive system: Identification of amonafide and BX795 as potential therapeutics via integrative Omics analysis

Weidong Liu[1☉], Jiaying Gao[2,3☉], Shuqiang Ren[1], Buhe Nashun[2,3], Fei Gao [1]*

**1** Genome Analysis Laboratory of the Ministry of Agriculture and Rural Affairs, Agricultural Genomics Institute at Shenzhen, Chinese Academy of Agricultural Sciences, Shenzhen, China, **2** Inner Mongolia Key Laboratory for Molecular Regulation of the Cell, Inner Mongolia University, Hohhot, China, **3** State Key Laboratory of Reproductive Regulation and Breeding of Grassland Livestock, School of Life Sciences, Inner Mongolia University, Hohhot, China

☉ These authors contributed equally to this work.
* flys828@gmail.com (FG)

## Abstract

### Background

Digestive system cancers, including esophageal, gastric, colorectal, pancreatic, hepatocellular, and biliary tract cancers, constitute a major global health challenge. Despite therapeutic advancements, prognosis remains poor, highlighting the urgent need for novel treatment strategies. We hypothesized that drug repositioning, facilitated by pan-cancer analyses, could lead to the identification of effective treatment strategies for these cancers.

### Results

We performed a comprehensive gene expression profiling of six major digestive system cancer types using The Cancer Genome Atlas data. Through integrative omics analysis, we identified 9,978 shared differentially expressed genes (DEGs) between colorectal cancer (CRC) and liver hepatocellular carcinoma (LIHC). Functional enrichment analysis revealed nine common Kyoto Encyclopedia of Genes and Genomes (KEGG) pathways, with Cell Cycle being a significant shared pathway. Protein-protein interaction (PPI) network analysis identified core genes within these pathways, including CCNE1, CHEK1, NXF1, NCBP2, and RPS27A. A Connectivity Map (CMap) analysis matched 1,147 small molecules, leading to the identification of Amonafide and BX795 as top candidates. These two drugs were validated and shown to inhibit the proliferation and migration of CRC (HT-29) and LIHC (HepG2) cells and induce cell cycle arrest and apoptosis.

**Data availability statement:** The Cancer Genome Atlas (TCGA) (https://portal.gdc. cancer.gov/) database was explored to obtain transcriptomic data for 6 types of digestive system cancers. A total of 316 samples were obtained, including 8 pairs of cholangio-carcinoma (CHOL), 50 pairs of colorectal cancer (CRC, including both colon and rectal adenocarcinoma (COAD/READ)), 13 pairs of esophageal carcinoma (ESCA), 50 pairs of liver hepatocellular carcinoma (LIHC), 4 pairs of pancreatic adenocarcinoma (PAAD), and 33 pairs of stomach adenocarcinoma (STAD) as well as their adjacent normal controls (S1 Table in Supporting Information).

**Funding:** This study was supported by the National Natural Science Foundation of China (32160145) and 'Leading the Charge with Open Competition' Project of Inner Mongolia (2022JBGS0021). This study was supported by the Basic Research Center, Innovation Program of Chinese Academy of Agricultural Sciences (No. CAAS-BRC-FNH-2025-02).

**Competing interests:** The authors have declared that no competing interests exist.

**Abbreviation:** CRC, colorectal cancer; CHOL, cholangiocarcinoma; ESCA, esophageal carcinoma; STAD, stomach adenocarcinoma; PAAD, pancreatic adenocarcinoma; LIHC, liver hepatocellular carcinoma; COAD/READ, colon and rectal adenocarcinoma; PPI, protein-protein interaction; GSEA, gene set enrichment analysis; RRHO, rank-rank hypergeometric overlap; KEGG, Kyoto encyclopedia of genes and genomes; WGCNA, weighted gene co-expression network analysis; FC, fold change; ME, module eigengenes; TOM, topological mverlap matrix; DEGs, differentially expressed genes; TPM, transcripts per kilobase <illion; UMAP, uniform manifold approximation and projection; CMap, connectivity map; TCGA, the cancer genome atlas; NCG, network of cancer genes& healthy drivers; CCK-8, cell counting kit-8; MOA, mechanisms of action.

## Conclusion

Our study demonstrates the utility of drug repositioning for identifying potential therapeutics for digestive system cancers. Amonafide and BX795 emerged as promising candidates in targeting both CRC and LIHC. Further in vivo studies and clinical trials are warranted to validate these findings.

## Background

Cancers of the digestive system, including esophageal, gastric, colorectal, pancreatic, hepatocellular, and biliary tract cancers, pose a significant global health burden. In recent years, drug development for digestive tract tumors has made notable progress, including advances in targeted therapies, immunotherapies, and chemotherapeutic agents [1,2]. Pembrolizumab is one of the most extensively studied immune checkpoint inhibitors for metastatic gastric cancer and is the first and only such inhibitor approved by the U.S. Food and Drug Administration (FDA) for gastric cancer treatment [3]. Meanwhile, TAS-102, an orally administered and well-tolerated chemotherapeutic agent, contains trifluridine as its active antitumor componen [4]. TAS-102 has been approved for the treatment of metastatic colorectal cancer and has demonstrated promising results [5]. Importantly, capecitabine, a prodrug of 5-FU has anti-tumor efficacy not only in CRC but also has demonstrated a good anti-tumor efficacy in hepatocellular carcinoma patients failing sorafenib therap.

Despite these advancements, drug development for digestive tract tumors continues to face numerous challenges. Most drugs are effective only for specific genetic mutations or protein expression profiles in particular cancers, limiting the patient population that benefits from them. This highlights the therapeutic challenges posed by tumor heterogeneity and drives researchers to explore more precise and effective treatment strategies, including drug repositioning.

Drug repositioning, the process of identifying new therapeutic uses for existing drugs, offers a promising approach to accelerate drug discovery and development [3,4]. By leveraging the extensive knowledge base and safety profiles of approved drugs, researchers can potentially identify new treatments for diseases, such as digestive system cancers, with greater efficiency and reduced risk [5]. However, successful drug repositioning requires a deep understanding of the molecular mechanisms underlying these diseases.

Pan-cancer analyses have emerged as a powerful tool for identifying shared molecular alterations across different cancer types [6,7]. By comparing the genomic and transcriptomic profiles of multiple cancer types, researchers can uncover common pathways and potential drug targets [8]. This approach has the potential to reveal novel therapeutic strategies that can be applied to a broader range of patients. Moreover, the integration of large-scale datasets with advanced computational methods has enabled the development of sophisticated algorithms and databases, such as the Connectivity Map (CMap), which facilitate the identification of drug candidates based on their ability to reverse disease-associated gene expression profiles [9].

Indeed, while digestive system cancers may share certain molecular mechanisms due to their common tissue origin, the extent to which these mechanisms are shared among different types of cancers and whether specific cancers have more similar mechanisms than others remains an open question that needs to be evaluated and addressed. In this study, we leverage the power of pan-cancer analyses and data-driven drug discovery to identify novel therapeutic strategies for digestive system cancers. By comparing the gene expression profiles of six different types of digestive system cancers, we aim to identify shared molecular mechanisms and potential drug targets. Our hypothesis is that by uncovering common pathways and molecular alterations, we can identify drug candidates that can effectively inhibit the growth and proliferation of multiple cancer types. Furthermore, we validated these drug candidates using in vitro cellular models to assess their therapeutic efficacy and safety.

## Methods

### Collection and processing of public transcriptomic data

The Cancer Genome Atlas (TCGA) (https://portal.gdc.cancer.gov/) database was explored to obtain transcriptomic data for 6 types of digestive system cancers. A total of 316 samples were obtained, including 8 pairs of cholangiocarcinoma (CHOL), 50 pairs of colorectal cancer (CRC, including both coloo and rectal adenocarcinoma (COAD/READ)), 13 pairs of esophageal carcinoma (ESCA), 50 pairs of liver hepatocellular carcinoma (LIHC), 4 pairs of pancreatic adenocarcinoma (PAAD), and 33 pairs of stomach adenocarcinoma (STAD) as well as their adjacent normal controls (S1 Table). We calculated gene expression levels using the transcripts per kilobase million (TPM) matrix, which was subsequently used for further data processing [10]. A filtering criterion was applied to identify genes with an expression level of 20 or above in at least 180 samples, yielding a dataset of 17,673 genes. Differential gene expression was assessed with DESeq2 using the raw count tables generated by TCGA (The Cancer Genome Atlas homepage.http://cancergenome.nih.gov/abouttcga.). Gene expression fold changes were computed, followed by dimensionality reduction via Uniform Manifold Approximation and Projection (UMAP) and hierarchical clustering.

### Rank-rank hypergeometric overlap analysis

To compare the global transcriptomic signatures of each COADREAD group and LIHC group, we applied the RRHO algorithm that is an unbiased and threshold-free method [11]. For each group, genes list were ranked according to the log 2-transformed fold-change generated by DESeq2, from the most up- to the most down-regulated ones. A hypergeometric test was performed to assess the significance of the similarity of gene profile, using a sliding window with step size (i.e., default) for each pair of impacts by using the RRHO2 (v1.0) package. A False Discovery Rate (FDR) correction was applied to adjust for the multiple hypothesis testing. The visualization of the output of this analysis was the RRHO level map, in which the most significant hypergeometric p-value (log 10 transformed and direction-signed) was labeled after computing all possible rank combinations, generating an index of the matrix for the most significant rank combination in each pair of impacts. Heatmaps generated using RRHO2 have top-right (both decreasing) and bottom-left (both increasing) quadrants, representing the concordant mRNA changes, while the top-left and bottom-right quadrants represent discordant overlaps (opposite directional overlap between datasets). For each comparison, the stratified RRHO method was used on log 10 (P-values) with the default step size for each quadrant, and adjusted Benjamini-Yekuteli p-values were calculated to determine significant concordant and discordant overlaps.

### Weighted gene co-expression network analysis

The R package "WGCNA" was utilized to construct gene co-expression networks for CRC and LIHC samples [12]. We first created an adjacency matrix of 9978 gene expressions using a soft threshold of 13 based on gene-gene correlation, which was then transformed into a topological overlap matrix (TOM). Hierarchical clustering of genes identified 18 co-expression

modules. We calculated module eigengenes (ME) for each co-expression module and correlated these with a binary variable representing the two cancer types, where CRC was assigned a value of 1 and LIHC a value of 0. The Pearson correlation coefficients indicated the degree of association between module expression patterns and the cancer type, with positive and negative values indicating a stronger association with CRC and LIHC, respectively.

### Functional enrichment analysis and PPI network analysis

Functional enrichment analysis was conducted using the clusterProfiler package (v4.0.5). Significantly enriched Kyoto Encyclopedia of Genes and Genomes (KEGG) pathways were identified using the enrichKEGG function. Gene set enrichment analysis (GSEA) was performed on a gene list sorted by log2 fold change (FC), utilizing the gseKEGG function, with a significance threshold set at $P < 0.05$ [13].

For protein-protein interaction (PPI) network construction, we used the STRING database (https://string-db.org/). Clustering analysis was performed using MCL clustering with default parameters, and TSV files for clusters 1, 2, and 3 were downloaded. Finally, Cytoscape (version 3.7.1) was employed to visualize the PPI networks [14]. In the visualization of PPI network, we defined connections between proteins at a correlation threshold of 0.7, indicating significant interactions. Node connectivity, based on the number of edges, was visually represented by color-coding: highly connected nodes were marked in red, while those with fewer connections were green.

### C-Map analysis

The CMap database is the largest perturbation-driven gene expression dataset, serving as a resource to explore relationships between diseases, genes, and therapeutics (https://clue.io/) [4]. In this study, potential drug candidates were screened through CMap analysis using 300 candidate central genes (150 up-regulated and 150 down-regulated) as input. The selected query parameters were "Gene expression (L1000)" and "Latest."

Matching results were filtered to include "Compound- trt_cp", "Knock Down- trt_kd" and "Over Expression- trt_sh" with a screening criterion of a score $\leq -1.2$ in CRC cell lines (HT29, HCT116, SW480, and NCIH508) and LIHC cell lines (HEPG2 and HUH7). Resuts of "Over Expression- trt_sh" were neglected later due to limited output. Positive scores indicated synergistic effects, while negative scores indicated antagonistic effects in cancer contexts.

### Statistical analysis

The P-value was calculated using the unpaired two-tailed Student's *t* test, and $P < 0.05$ was considered statistically significant. Hierarchical clustering analysis was performed using pheatmap (v1.0.12) package with Pearson correlation. For RNA-seq analysis, significant differentially expressed genes (DEGs) with fold change > 2 and Benjamini-Hochberg (BH)–adjusted P-value < 0.01 were identified by DESeq2 and adjusted for litter. The clusterProfiler package was used to determine potential KEGG. Terms with Benjamini-Hochberg corrected P-value < 0.05 were defined as significantly enriched terms and pathways. All the statistical analyses were done using available packages in R (v4.1.3).

### Cell culture

The human colon cancer cell line HT29 and the human liver cancer cell line HepG2 (both purchased from ATCC in the United States) were cultured in DMEM medium (C11995500BT, Gibco, USA) supplemented with 10% fetal bovine serum (FSP500, ExCell Bio, China) and 1% penicillin-streptomycin (C3420-0100, Vivacell, China). The cells were incubated at 37°C in 5% carbon dioxide.

### Cell viability assay

The effects of Amonafide and BX795 on the viability of HT29 and HepG2 cells were evaluated using the CCK8 kit (MI00615A, MISHU, China). Amonafide (HY-10982) and BX795 (HY-10514) were purchased from Shanghai

MedChemExpress Bio-Technology Co., Ltd. The drugs were diluted with DMEM medium to achieve the desired concentrations. HT29 and HepG2 cells were seeded in 96-well plates at a density of $1 \times 10^4$ cells per well and cultured in a 5% CO2 incubator at 37°C for 24 hours to allow cell adhesion. The culture medium was then replaced with DMEM medium containing various concentrations of Amonafide and BX795 (0, 5, 10, 20 μM). After 12 hours, each well was treated with 10 μL of CCK8 reagent and incubated for 1 hour. Absorbance at 450 nm was measured using an enzyme-linked immunosorbent assay (ELISA) reader (SpectraMax iD5, Shanghai, USA). The same procedure was repeated for 24, 36, and 48 hours.

## Colony formation

HepG2 cells were seeded in 6-well plates at a density of $3 \times 10^3$ cells per well, and HT29 cells were seeded at a density of $8 \times 10^3$ cells per well, with three replicates for each group. The cells were incubated at 37°C and 5% CO2 for 24 hours to allow cell adhesion, and the culture medium was discarded. Amonafide and BX795 were prepared in DMEM containing 10% serum at various concentrations (0, 0.1, 1, 5 μM), and the cells were cultured for 48 hours, with the medium being changed once. After visible colonies formed, the cells were washed with PBS and fixed at room temperature for 20 minutes in 4% tissue cell fixative solution. After PBS washing, they were stained with 0.5% crystal violet solution for 15 minutes, washed again with PBS, dried, and observed under a microscope. Colonies were counted using ImageJ software.

## Cell scratch assay

HT29 and HepG2 cells were seeded in six-well plates with three replicates per group. Once the cells reached 80–90% confluence, the culture medium was removed. Amonafide and BX795 were prepared in DMEM containing 1% fetal bovine serum, and the cells were cultured at different concentrations (0, 1, 5, 10 μM). A scratch was made using a sterile 200 μL pipette tip, and PBS was used to remove cell debris. Images were taken with an inverted microscope to record the scratch location. The wounds were then incubated at 37°C in 5% CO2 for 48 hours, and photographs were taken with an optical inverted microscope connected to a real-time imaging system. Images from three separate experiments were recorded for each group, and cell migration distances were analyzed using ImageJ.

## Flow cytometry

A cell cycle detection kit (C6031S, UElandy, China) was used to assess the effects of Amonafide and BX795 on the cell cycle of HT29 and HepG2 cells. Cells were cultured in 60 mm dishes and treated with Amonafide and BX795 at different concentrations (0, 10 μM) for 24 hours. The cells were collected, centrifuged, and washed with 1 mL of pre-cooled PBS. Subsequently, 1 mL of 75% ethanol was added, and the cells were fixed overnight at −20°C. After centrifugation to remove the supernatant, the cells were washed with pre-cooled PBS. Appropriate propidium iodide staining solution was added, and cells were incubated at room temperature in the dark for 15–30 minutes. Cell cycle analysis was conducted using a flow cytometer (Beckman, USA).

## Apoptosis assessment

Apoptosis was assessed using the Annexin V-FITC/PI staining kit (FA101, TransGen Biotech, China). HT29 and HepG2 cells were cultured in 60 mm dishes and treated with Amonafide and BX795 at different concentrations (0, 10 μM) for 24 hours. Cells were collected, centrifuged, washed with pre-cooled PBS, and resuspended in 100 μL of pre-cooled 1×Annexin V Binding Buffer. Cells were then incubated with 5 μL of Annexin V-FITC and 5UL-PI in the dark for 15 minutes at room temperature. After adding 400 μL of pre-cooled 1×Annexin V Binding Buffer, the samples were placed on ice and analyzed by flow cytometry.

## Results

### Assessment of gene expression alterations across six digestive system cancers

Gene expression analysis reveals the molecular signatures of cancers, potentially highlighting shared therapeutic targets and biomarkers. In order to delineate common and distinct molecular features, we collected paired transcriptome data (cancerous and adjacent non-cancerous tissues) for six digestive system cancer types, amounting to 316 samples (Methods, S1 Table). After dimensionality reduction with Uniform Manifold Approximation and Projection (UMAP), the samples from six cancer types of the digestive system did not exhibit significant differences and were uniformly distributed, indicating that they share similar features in their gene expression profiles (Fig 1A). In line with the UMAP results, hierarchical clustering analysis also showed no distinct clusters among samples from various cancer types, presenting a homogeneous distribution (Fig 1B).

Further analysis focused on identifying differentially expressed genes (DEGs) involved in distinguishing between six types of cancers, and included the calculation of intersections among different combinations of DEGs (S1A Fig, S2 Table). The combination of cholangiocarcinoma (CHOL) and liver hepatocellular carcinoma (LIHC) exhibited the highest number of shared DEGs, with 738 common genes, while colorectal cancer (CRC) and LIHC shared 358 DEGs, ranking sixth. The combination of all six cancer types showed the least number of shared DEGs, totaling only 4 genes. As the number of cancer combinations increased, the number of shared DEGs decreased. Thus, while there is overall similarity in gene expression profiles among various cancer types, specific combinations of cancers exhibit higher similarity in characteristic DEGs.

To assess the functional pathway similarities of DEGs across different cancer types, we also evaluated the intersections of Kyoto Encyclopedia of Genes and Genomes (KEGG) pathways among various combinations of DEGs (S1B Fig, S3 Table). The combination of CHOL and CRC demonstrated the highest number of shared KEGG pathways, with 26 common pathways, while colorectal cancer and liver hepatocellular carcinoma shared 16 pathways. The combination of all

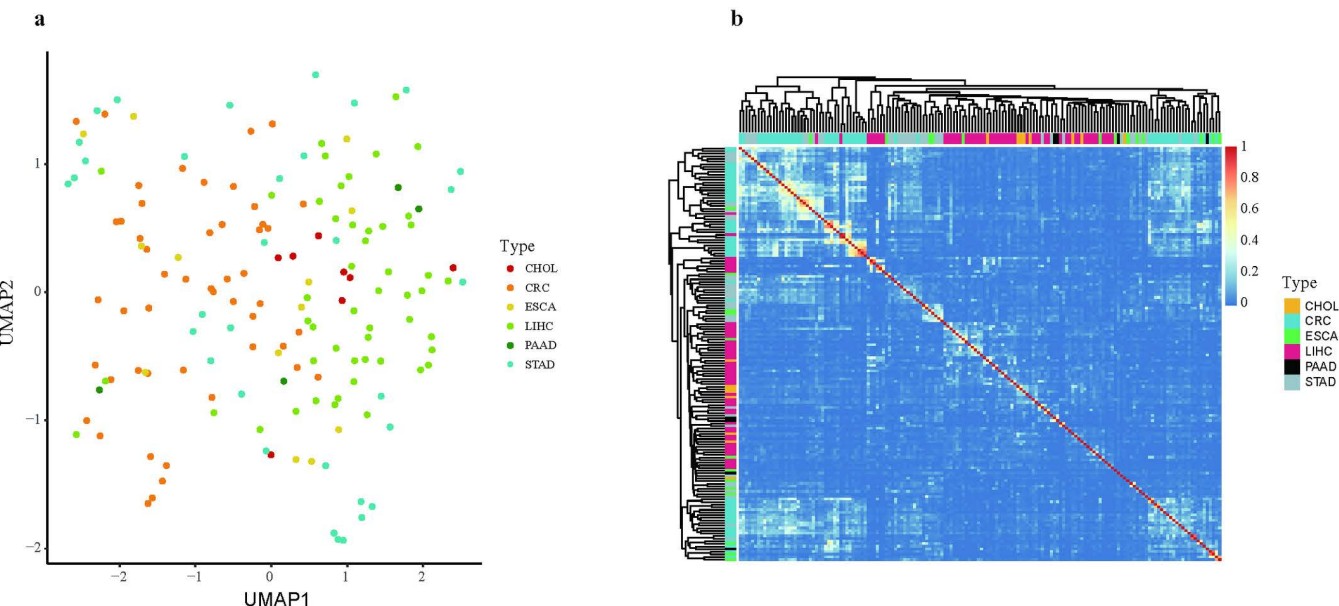

**Fig 1. Gene Expression Correlation and Dimensionality Reduction Analysis in Digestive System Cancers. (a)** Uniform Manifold Approximation and Projection (UMAP) analysis visualizes the distribution of fold change (FC) in pairwise samples. **(b)** Heatmap depicts the Pearson correlation coefficients between gene expression profiles of different cancer samples. 8 cholangiocarcinoma (CHOL), 50 colorectal cancer (CRC), 13 esophageal carcinoma (ESCA), 50 liver hepatocellular carcinoma (LIHC), 4 pancreatic adenocarcinoma (PAAD) and 33 stomach adenocarcinoma (STAD) samples were used.

six cancer types exhibited the fewest shared pathways, with only 4 common pathways, including Cell Cycle and Cytokine-cytokine receptor interaction.

## Comprehensive C-Map analysis

To further determine which combinations of cancers share more similar characteristics and help decipher common features across multiple tumors, we conducted Connectivity Map (C-Map) matching analysis. The top 150 upregulated and downregulated DEGs from each of the six tumor types were input into CMap for small molecule matching based on similar characteristics. All the cell lines were applied in this process. Matching scores for the same drug treated by the respective cancer type cell lines were obtained by combining the median scores into a unique score. The matching drugs across different cancer types were overlapped in pairs to generate a list of co-acting drugs (see Method). The results indicate that the liver-intestinal interaction involves the highest number of small molecules, totaling 1147 (Fig 2A), and the highest absolute values of small molecule matching scores. It is important to note that due to the lack of CMap data for CHOL and ESCA cell lines, these two cancer types were excluded from the analysis.

The CMap analyses also revealed genes that upon knockdown (KD) led to transcriptional signitures similar to the input DEG set of CRC or LIHC. Based on a criterion of norm_cs less than 1.2, we obtained 588 and 579 genes for CRC and LIHC, respectively (S4 Table). KEGG analysis indicated that enrichment of these two list of genes showed very similar pathways, expecially PI3K-Akt and MAPK signaling pathways (Fig 2B and C, S2 Fig). Therefore, we consider the liver-intestinal interaction to be more closely associated and have chosen it as the focus for subsequent research. Building upon the DEG analysis and C-Map-based drug repurposing findings from six digestive tract cancers, we have narrowed our focus to an in-depth investigation of CRC and LIHC.

## Identification of shared core gene expression modules between CRC and LIHC

Next, we employed an alternative analytical strategy to further explore the commonalities in the genetic pathways between CRC and LIHC. Initially, we utilized the Rank Rank Hypergeometric Overlap (RRHO) algorithm to compare the global

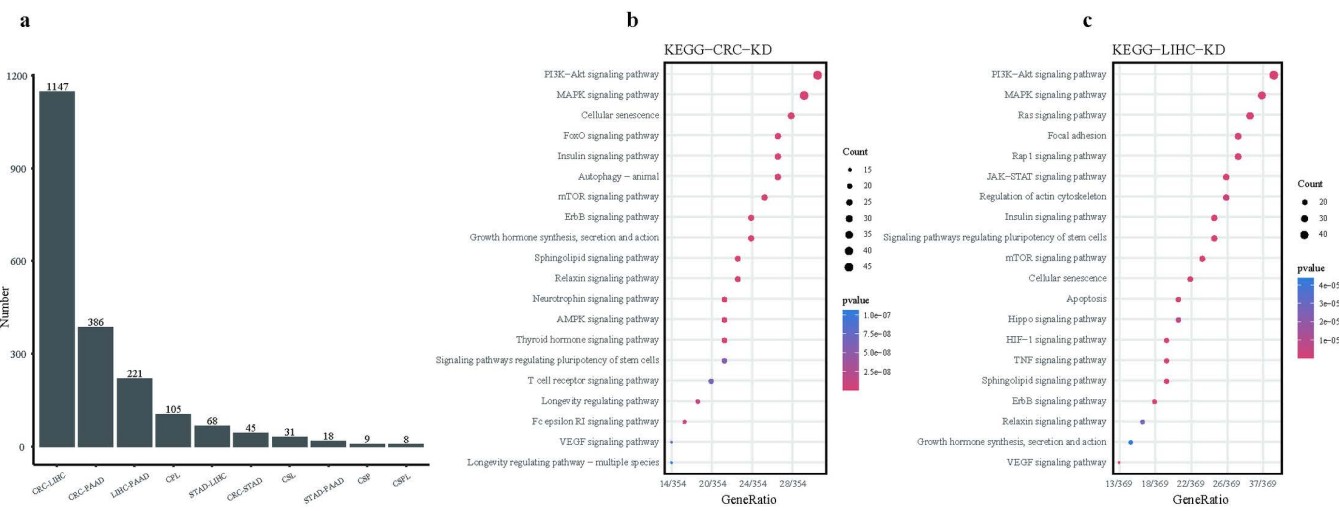

**Fig 2. Post-CMap Matching Analysis of Compounds and Genes. (a)** Bar diagram representation of the intersection of CMap compounds matched to four types of cancer, quantifying the shared compounds among two, three, and four cancer types. **(b)** Kyoto Encyclopedia of Genes and Genomes (KEGG) enrichment pathways of CMap knockdown genes matched to differentially expressed genes (DEGs) in CRC. **(c)** KEGG enrichment pathways of CMap knockdown genes matched to DEGs in LIHC.

gene expression profiles of CRC and LIHC, which contains 17673 filtered genes (Methods). The heatmap revealed that the gene expression signatures were predominantly concentrated in the first and third quadrants, indicating a generally concordant expression trend (Fig 3A). Among these genes with similar expression trends, we identified 5,415 upregulated and 4,564 downregulated genes, totaling 9,978 genes ($p < 0.05$), which were then subjected to Weighted Gene Co-expression Network Analysis (WGCNA).

The WGCNA identified 18 distinct each co-expression modules, for each of which we calculated module eigengenes (ME) and correlated them with a binary variable representing the two cancer types. We then ranked the co-expression modules based on their correlation coefficients, aiming to identify modules with high relevance to both CRC and LIHC. The results highlighted the turquoise and blue modules, which were enriched with 2,929 and 1,189 genes, respectively. Notably, these two modules exhibited a strong correlation with the gene expression patterns of both cancer types (Fig 3B). Pathway analysis of these gene sets revealed nine common pathways ($p < 0.05$): Cell Cycle, Nucleocytoplasmic Transport, Ubiquitin Mediated Proteolysis, ATP-dependent Chromatin Remodeling, DNA Replication, RNA Polymerase, Base Excision Repair, Non-homologous End-joining, and Endocytosis (S5 Table). A total of 328 genes were found to be enriched in these pathways, suggesting their potential as core similar gene expression modules in these two types of cancer (S5 Table).

We then combined the shared 358 DEGs between CRC and LIHC (S2 Table) and these 328 genes for protein-protein interaction (PPI) analysis using the STRING database. The gene clusters were categorized into three primary groups, each comprising 86, 37, and 36 proteins, respectively. Cluster 1 was primarily enriched in the Cell Cycle pathway, with 49 out of 86 proteins; Cluster 2 in Ubiquitin Mediated Proteolysis pathway, with 33 out of 37 proteins; and Cluster 3 in RNA Translation pathway, with 32 out of 36 proteins (S6 Table). We also searched in four databases intOGEN, OncokB, Cancer Gene Census, and Network of Cancer Genes& Healthy Drivers (NCG) to identify driver genes of these two cancers (S7 Table) and matached with these DEGs. Using Cytoscape, we then visualized the core proteins of these clusters and annotated the driver genes in the liver and intestine. In the Cell Cycle protein network (Fig 3C), CCNE1 and CHEK1 emerged as core proteins, which were also identified as driver genes in both the CRC and LIHC. Similarly, the core proteins of NXF1 and NCBP2 in the Ubiquitin Mediated Proteolysis pathway (Fig 3D), and RPS27A as a core protein in RNA Translation pathway (Fig 3E), were also drive genes of the two cancers. These pathways may hold the potential for targeted therapeutic intervention that leverages the shared molecular mechanisms of CRC and LIHC.

## Selection of repositioned key drug candidates commonly applicable for both CRC and LIHC

Following the identification of shared differentially expressed genes (DEGs) and functional pathways in liver-intestine tumors, we initiated a stringent drug candidate screening process aimed at identifying potential treatments for both CRC and LIHC. This process was informed by preliminary screening results, leveraging data from the Connectivity Map (CMap) database, which encompasses a broad spectrum of cell lines (S3 Fig).

Our approach involved a refined selection of compounds based on their expression profiles in specific cell types. For LIHC analysis, we focused on HEPG2 and HUH7 cell lines, identifying 5,972 compounds. In the case of CRC, we considered HCT116, HT29, SW480, and NCIH508 cell lines, yielding 11,282 compounds. The intersection of these drug lists revealed 1,147 compounds common to both CRC and LIHC.

Given the variability in scores associated with these drugs for the two different cancers, we calculated the median value of their scores and re-ranked the drugs accordingly. This step was crucial for prioritizing compounds with a median score of $\leq -1.2$, which indicated a significant potential for therapeutic intervention. Subsequently, we filtered our list to include only those compounds with documented mechanisms of action (MOA), and conducted an in-depth investigation of each compound, which encompassed the examination of compound names, drug types, and therapeutic indications from FDA, CMap, DrugBank, and PubMed databases. Further, information of driver genes from four databases intOGEN, OncokB, Cancer Gene Census, and Network of Cancer Genes& Healthy Drivers (NCG) were also included (S7 Table).

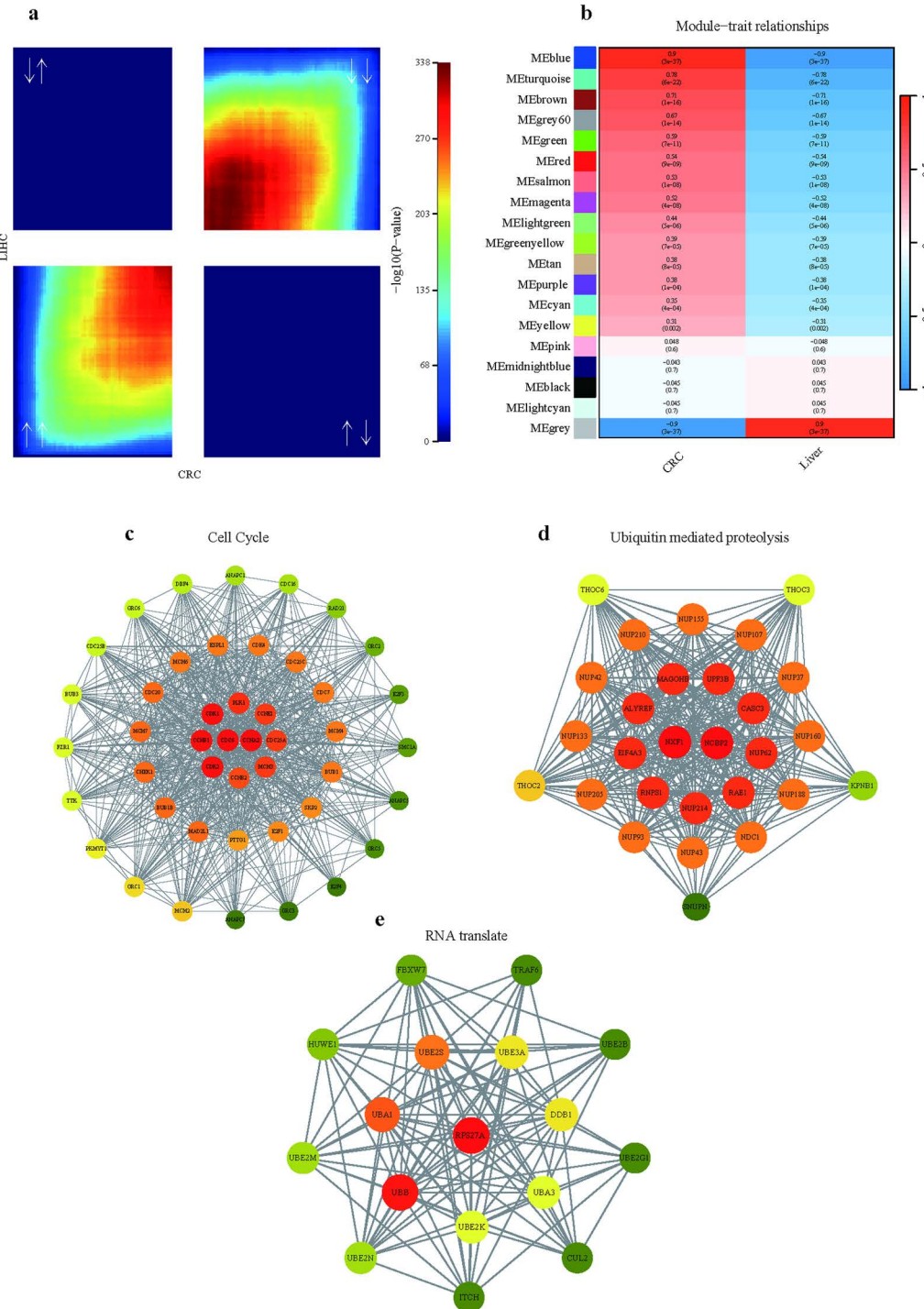

**Fig 3. Identification of Shared Core Gene Expression Modules between CRC and Liver Cancer. (a)** Rank-rank hypergeometric overlap (RRHO) heatmaps reveal significant overlaps in gene expression changes between CRC and liver cancer (P value < 0.05), with white arrows indicating up- or downregulation. **(b)** Weighted Gene Co-expression Network Analysis (WGCNA) heatmap displays co-expression modules. Pearson's correlation coefficients and their respective p-values are indicated. **(c)** Protein-protein interaction (PPI) network enriched for the Cell Cycle pathway within cluster 1, highlighting genes with high (red) and low (green) correlations. **(d)** PPI network for Ubiquitin mediated proteolysis within cluster 2, with color coding for correlation strength. **(e)** PPI network for RNA translation within cluster 3, using the same color coding scheme as in panels (c) and **(d)**.

The candidates were chosen for their potential to modulate the expression of shared DEGs and their involvement in key pathways, highlighting their repositioning potential for the treatment of liver and colorectal cancers. The selection process was meticulous, ensuring that the candidates not only aligned with our multi-omics analysis but also possessed a well-documented safety profile and demonstrated therapeutic relevance. Through this rigorous screening process, we established a prioritized list of core drugs, with Amonafide and BX795 emerging as the top two candidates (S8 Table). These drugs were selected for their potential to target the molecular underpinnings of CRC and LIHC, offering a promising avenue for further research and development.

### Inhibition of CRC and LIHC cell proliferation and migration by Amonafide and BX795

In our pursuit to identify drugs capable of modulating the behavior of CRC and LIHC cells, we focused on the top two candidates in the list, i.e., Amonafide and BX795. To validate the inhibitory effects of Amonafide and BX795 on the proliferation of CRC (HT-29) and LIHC (HepG2), these cells were evaluated using the Cell Counting Kit-8 (CCK-8) assay. The results indicated a significant reduction in cell viability following treatment with both drugs, with a dose-dependent decrease in proliferation rates (Fig 4A). This was corroborated by colony formation assays, which revealed a marked reduction in the number of colonies formed by treated cells compared to controls (Fig 4B, S4 Fig), underscoring the drugs' ability to suppress cell clonogenic potential.

Futher, the impact of Amonafide and BX795 on cell migration was assessed using a scratch assay. The results demonstrated a significant impairment in the migration capacity of both HT-29 and HepG2 cells post-treatment, with a noticeable

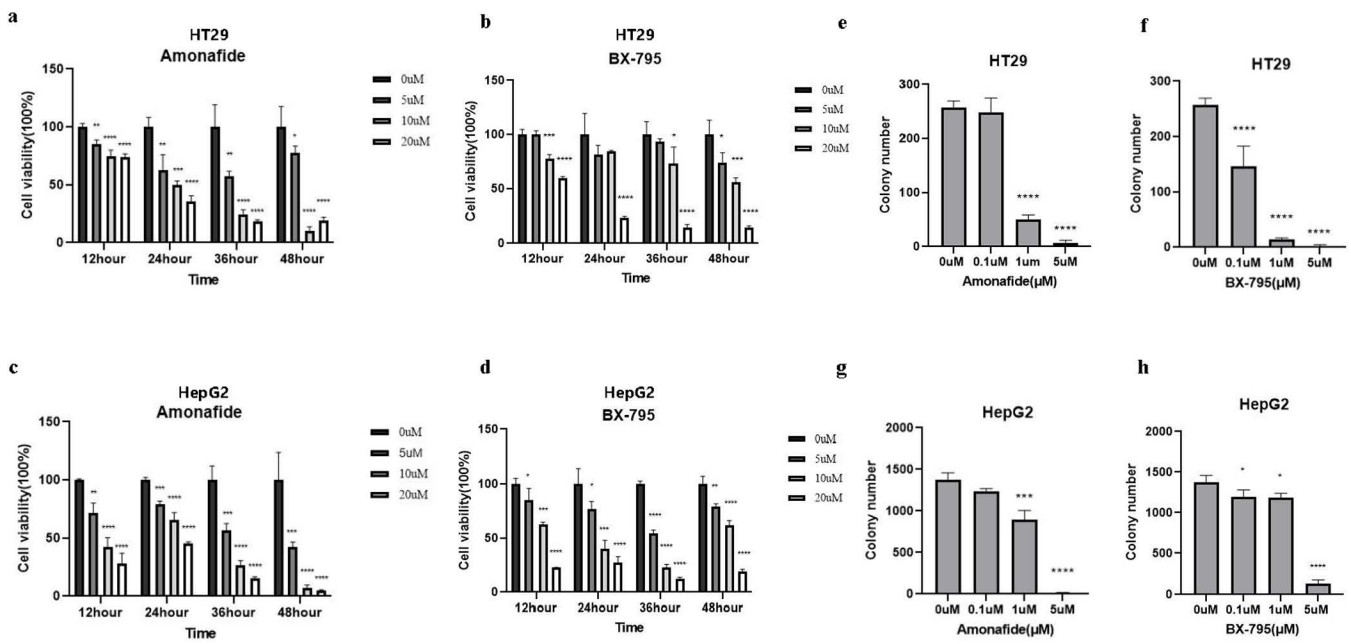

**Fig 4. Inhibitory Effects of Amonafide and BX795 on Cell Viability and Clonogenicity in CRC and LIHC Cells. (a)** CCK8 assay shows the time-dependent impact of Amonafide on HT29 cell viability. **(b)** CCK8 assay reveals the effect of BX795 on HT29 cell viability over time. **(c)** Amonafide's influence on HepG2 cell viability as determined by CCK8 assay. **(d)** BX795's effect on HepG2 cell viability as measured by CCK8 assay. **(e)** Clonogenic ability of HT29 cells treated with Amonafide assessed by colony formation assay. **(f)** Colony formation assay for HT29 cells treated with BX795 to determine clonogenic potential. **(g)** Clonogenicity of HepG2 cells under Amonafide treatment measured by colony formation assay. **(h)** Colony formation assay shows the effect of BX795 on HepG2 cell clonogenicity. Data represents at least three independent experiments (n ≥ 3). Statistical significance is denoted by * for P < 0.05, ** for P < 0.01, *** for P < 0.001, and **** for P < 0.0001.

reduction in the distance migrated relative to control cells (Fig 2). This suggests that both drugs can impede the invasive capabilities of these cells.

## Validation of effects on cell cycle and apoptosis

Flow cytometry analysis was employed to examine the effects of Amonafide and BX795 on the cell cycle progression of HT-29 and HepG2 cells. Amonafide treatment led to an accumulation of HT-29 cells in the S phase, with a 1.88-fold increase in the proportion of cells in this phase. Conversely, BX795 induced a significant increase in the proportion of cells in the G2/M phase, with a 1.75-fold elevation in HT-29 cells and a 4.17-fold increase in HepG2 cells (Fig 5A and B). These findings indicate that Amonafide and BX795 can disrupt cell cycle progression at specific checkpoints. Further, annexin V-FITC/PI staining was utilized to assess apoptosis induction. Both Amonafide and BX795 significantly increased the proportion of apoptotic cells compared to the control group, with BX795 showing a particularly pronounced effect on inducing apoptosis in both cell types (Fig 6A and B). The data collectively demonstrate that Amonafide and BX795 possess potent inhibitory effects on the growth, proliferation, and invasion of CRC and LIHC cells. These drugs not only impede cell cycle progression and migration but also promote apoptosis, highlighting their therapeutic potential in the context of liver and intestinal malignancies.

## Discussion

The increasing global health burden of digestive system cancers necessitates the pursuit of innovative therapeutic strategies [15]. Traditional drug development is a time-consuming, costly, and high-risk process, requiring extensive testing at the cellular, animal model, and clinical levels to evaluate drug safety, side effects, and efficacy [8]. For complex diseases, most drugs prove insufficiently effective, and the success rate of drug discovery continues to decline. This is attributed to the low quality and reproducibility issues plaguing much of the foundational and preclinical research, as well as the limitations of the traditional "one disease, one target, one drug" approach, which hinders progress in cancer drug development.

Compared to de novo drug development, drug repurposing offers significant advantages. It has emerged as an attractive strategy for discovering new therapeutic applications for existing drugs, accelerating the drug development process [9]. Due to the complex interconnections between diseases, drugs originally developed for one condition may also prove effective against others [10]. In previous preclinical and clinical studies, drug repurposing has demonstrated favorable pharmacokinetic profiles and safety assessments, reducing the risks, time, and costs associated with drug development while providing more effective treatment options for cancer [11]. Leveraging existing pharmacokinetic and toxicological research, drug repurposing excels in cost-effectiveness, development time, and risk management, offering lower costs, shorter timelines, and reduced risks, thereby expediting therapeutic research [12].

Our study leverages the power of pan-cancer analyses and drug repositioning to identify novel treatment avenues for these challenging malignancies [7,16]. By conducting a comprehensive assessment of gene expression alterations across six major types of digestive system cancers, we have unveiled shared molecular mechanisms that could serve as potential drug targets. Our findings not only advance the understanding of these cancers' molecular landscapes but also provide a foundation for the development of targeted therapies.

To identify the core gene expression modules, we integrated the usage of the RRHO algorithm and WGCNA analysis, which is key outcome of our study. The discovery of 9,978 genes with concordant expression trends between CRC and LIHC, and the subsequent identification of key pathways enriched within these gene sets, provides a deeper understanding of the molecular underpinnings of these cancers [17,18]. The functional enrichment analysis and PPI network analysis provided further insights into the potential mechanisms of action of these drugs [19,20]. The significant overlap in KEGG pathways, particularly those involved in cell cycle regulation, underscores the importance of these processes in the pathogenesis of these cancers. Notably, core proteins CCNE1 and CHEK1 were also identified as driver genes within this pathway. This finding is consistent with existing knowledge of cancer biology, as the Cell Cycle pathway has long been

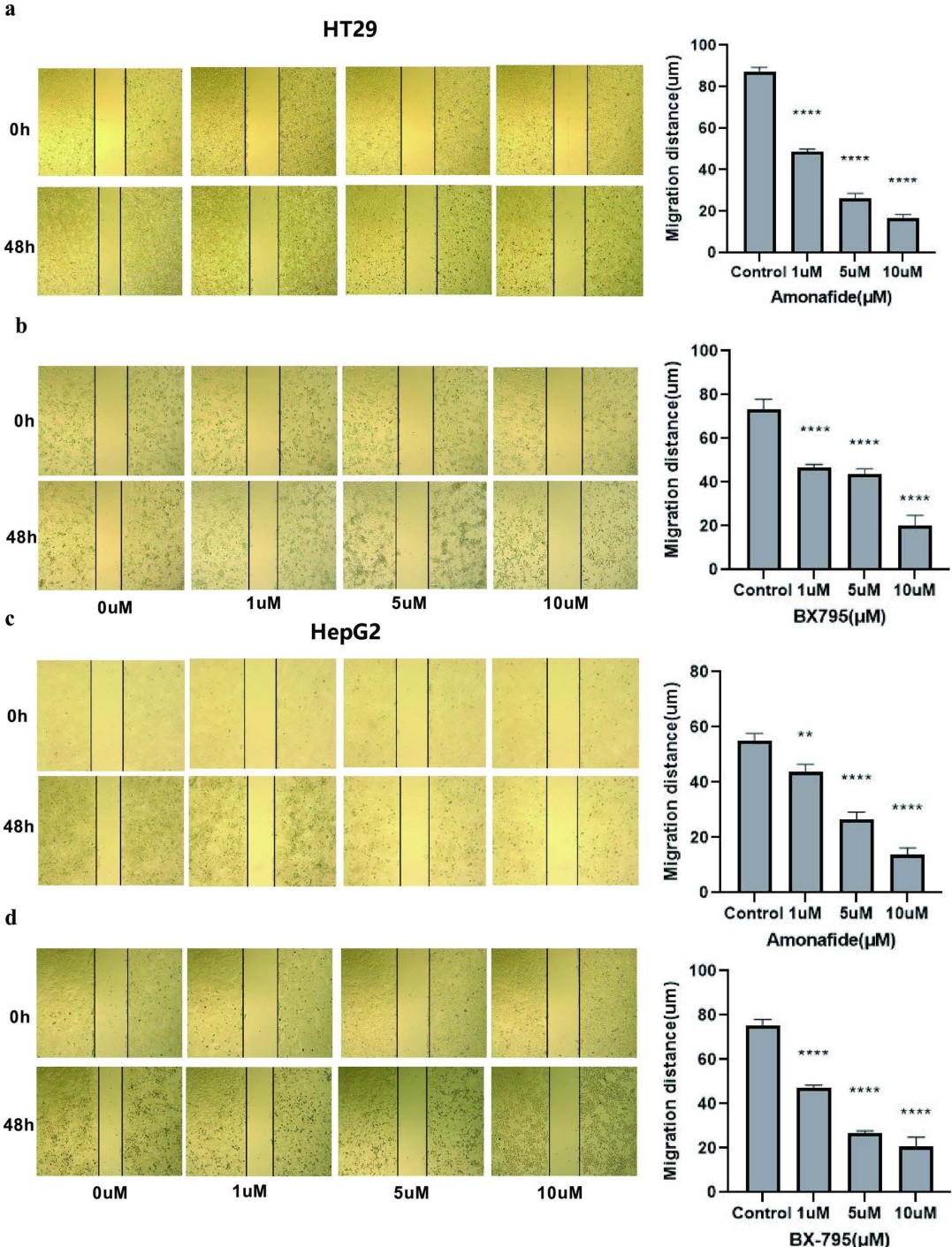

**Fig 5. Impact of Amonafide and BX795 on Cell Migration in HT29 and HepG2 Cells. (a)** Cell scratch assay displays Amonafide's effect on HT29 cell migration; left panel shows experimental results, and right panel shows statistical analysis of migration distance. **(b)** Cell scratch assay illustrates the impact of BX795 on HT29 cell migration; includes both experimental results and migration distance statistics. **(c)** Amonafide's influence on HepG2 cell migration as detected by cell scratch assay; presents both experimental outcome and distance quantification. **(d)** Cell scratch assay results and statistical analysis for HepG2 cell migration under BX795 treatment. Each experiment was repeated at least three times (n ≥ 3). Statistical significance indicated by * for P < 0.05, ** for P < 0.01, *** for P < 0.001, and **** for P < 0.0001.

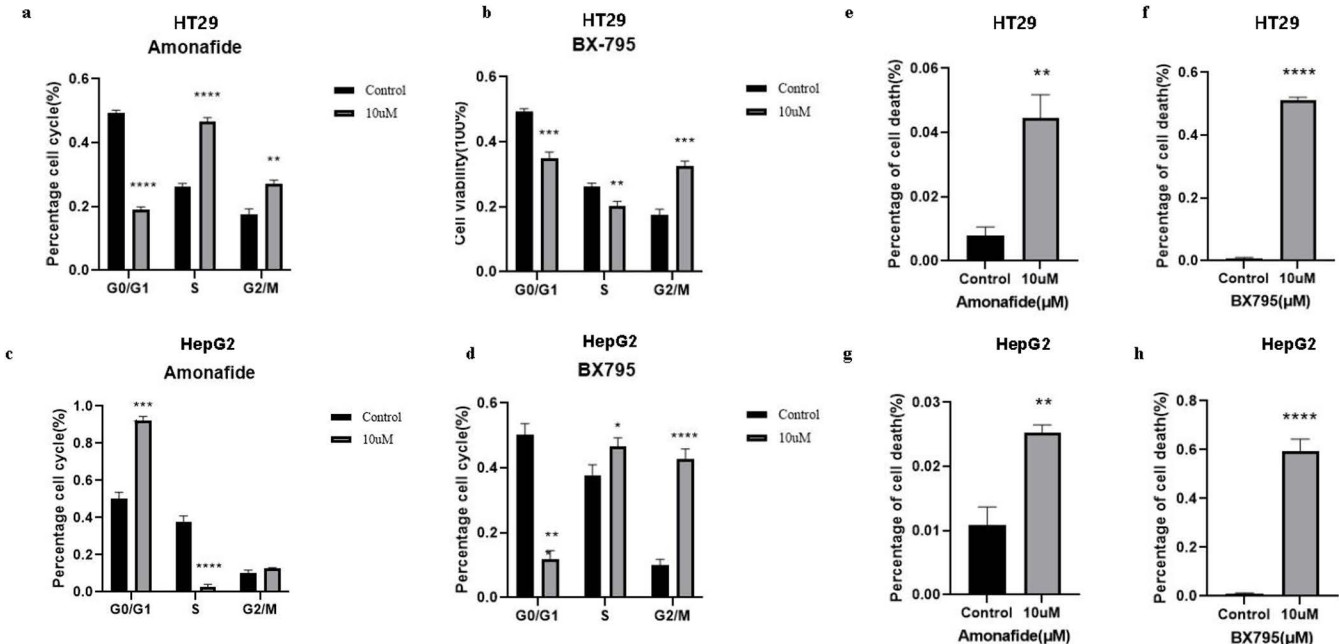

**Fig 6. Modulation of Cell Cycle and Induction of Apoptosis by Amonafide and BX795. (a)** The effect of Amonafide on the cell cycle progression of HT29 cells as measured by flow cytometry. **(b)** Flow cytometry analysis of HT29 cell cycle perturbation by BX795. **(c)** Amonafide's impact on HepG2 cell cycle as detected by flow cytometry. **(d)** Analysis of HepG2 cell cycle effects due to BX795 treatment using flow cytometry. **(e)** Apoptosis rate in HT29 cells as measured by Annexin V-FITC/PI flow cytometry. **(f)** Flow cytometric assessment of HT29 cell apoptosis induced by BX795. **(g)** Determination of HepG2 cell apoptosis rate by flow cytometry. **(h)** Impact of BX795 on HepG2 cell apoptosis as measured by flow cytometry. Experiments were conducted in triplicate (n ≥ 3). * denotes P < 0.05, ** for P < 0.01, *** for P < 0.001, and **** for P < 0.0001.

identified as a common functional pathway as a therapeutic target [21]. further validating our approach. In addition, Ubiquitin Mediated Proteolysis and RNA Translation were recognized as potential pathways for targeted therapies for both LIHC and CRC. The identification of core proteins NXF1 and NCBP2 as driver genes in this pathway provides new avenues for exploring the functional implications of the ubiquitin-proteasome system in cancer [22]. Similarly, RPS27A, a core protein identified in association with RNA Translation, could be pivotal in the control of translation initiation, which is a process frequently deregulated in cancer due to its impact on oncogene and tumor suppressor expression [23].

Based on these results, our study's approach to identify candidate compounds for development of targeted therapies is strengthened by the systematic and data-driven methodology [3,5,24]. By harnessing the Connectivity Map (CMap) and leveraging the extensive knowledge base of approved drugs, we have identified 1,147 small molecules that share similarities in their effects on gene expression profiles of CRC and LIHC, which is a testament to the power of large-scale datasets and advanced computational methods.

Our study further contributes to the field by demonstrating the potential of Amonafide [25] and BX795 [26] in targeting both CRC and LIHC. These drugs demonstrated the ability to inhibit cell viability, proliferation, and migration in CRC and LIHC cells, inducing cell cycle arrest and apoptosis. These effects are particularly noteworthy, as they highlight the potential of these agents to disrupt key oncogenic processes. While previous reports have suggested the use of Amonafide for treating multidrug-resistant colon cancer cells [27] and BX795 for patients with radiation resistant hepatocellular cancer [28], our research extends this knowledge by revealing the shared molecular pathways that may be targeted by Amonafide and BX795, offering a rationale for their use in a combined therapeutic strategy for CRC and LIHC. This knowledge may facilitate the development of more precise therapeutic approaches that leverage the known effects of Amonafide and

BX795. Our findings are also bolstered by the rigorous screening process employed in the selection of drug candidates. The prioritization of Amonafide and BX795 was based not only on their alignment with our multi-omics analysis but also on their documented safety profiles and therapeutic relevance [26,29–31]. This dual focus on efficacy and safety is critical in translating research findings into clinical practice.

However, our study is not without limitations. The *in vitro* nature of our experiments means that the effects of these drugs in a clinical setting remain to be established. Future *in vivo* studies and clinical trials will be essential to validate the therapeutic potential of Amonafide and BX795. Moreover, our research lacks Compare drug efficacy to standard therapies, but these drugs do have inhibitory effects on two types of cancers. Additionally, while our study provides insights into the potential mechanisms of action of these drugs [26,32], a more detailed understanding of these mechanisms will require further investigation. Despite these limitations, the identification of shared molecular mechanisms and the successful repositioning of drugs for these cancers may pave the way for more personalized and effective treatment strategies [29]. By targeting key pathways such as the Cell Cycle, Ubiquitin Mediated Proteolysis and RNA Translation, our approach has the potential to improve patient outcomes and address the pressing need for more effective therapies [33].

## Conclusions

Our study harnesses the power of pan-cancer analyses and drug repositioning to identify novel therapeutic strategies for digestive system cancers. The identification of Amonafide and BX795 as promising drug candidates, and the elucidation of shared molecular mechanisms, represent significant advancements. These findings provide a foundation for future research and these compounds are known clinical research drugs. Drug repurposing may change the treatment prospects for patients with these diseases.

## Supporting information

**S1 Fig. Bar diagram representation of the intersection of (a) DEGs and (b) their KEGG pathways matched to six types of cancer.**
(TIF)

**S2 Fig. Intersection statistics of CMap-matched knockdown genes for four cancer types.**
(TIF)

**S3 Fig. Flow chart of the screening process for LIHC-CRC common potential drugs.**
(TIF)

**S4 Fig. Cell cycle histograms of HT29 cells and HepG2 cells treated with 10uM concentration of Amonafide and BX795 for 24h.**
(TIF)

**S5 Fig. Histograms for flow cytometry demonstrating HT29 and HepG2 apoptosis after 24 h of treatment with 10uM Amonafide and BX795 and double stained with annexin V and PI.**
(TIF)

**S6 Fig. Images of colony formation tests after treatment with both cells by Amonafide and BX-795.**
(TIF)

**S1 Table. Gene expression values (transcripts per kilobase million (TPM) reprocessed from the transcriptomic data collected from TCGA.** The orginal datasets can be found via Project ID (TCGA cancer type indicated), Case and Sample ID (patient number indicated).
(XLSX)

**S2 Table. List of differentially expressed genes in six cancer types with details provided for CRC and LIHC.**
(XLSX)

**S3 Table. Results of KEGG analyses.**
(XLSX)

**S4 Table. List of genes that upon knockdown (KD) led to transcriptional signitures similar to the input DEG set of CRC or LIHC and their KEGG enrichment.**
(XLSX)

**S5 Table. Shared KEGG pathways between modules Meturquoise and Meblue, and the enriched gene sets within these common pathways.**
(XLSX)

**S6 Table. Protein-Protein Interaction Gene Sets and KEGG Pathway Results Post-STRING Clustering.**
(XLSX)

**S7 Table. Common driver genes of 6 cancers queried through 4 databases.**
(XLSX)

**S8 Table. List of 23 CMap-matched drugs common to CRC and LIHC.** In Target Name, the driver genes of CRC are marked in red; The driver genes of LIHC are marked in blue; The shared driver genes between CRC and LIHC are marked in green; The core genes from PPI results that are significantly enriched in the Cell Cycle pathway are marked in orange.
(XLSX)

## Acknowledgments

We thank Dr. XY for the guidance with C-Map and RRHO analyses.

## Author contributions

**Data curation:** Weidong Liu.

**Validation:** Jiaying Gao, Shuqiang Ren.

**Writing – original draft:** Weidong Liu, Buhe Nashun, Fei Gao.

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
