## [Decision Letter · Decision Letter 0]

13 Apr 2025

PONE-D-25-07116Drug Repositioning for Pan-Cancers of the Digestive System: Identification of Amonafide and BX795 as Potential Therapeutics via Integrative Omics AnalysisPLOS ONE

Dear Dr. Gao,

Thank you for submitting your manuscript to PLOS ONE. After careful consideration, we feel that it has merit but does not fully meet PLOS ONE’s publication criteria as it currently stands. Therefore, we invite you to submit a revised version of the manuscript that addresses the points raised during the review process.

We look forward to receiving your revised manuscript.

Kind regards,

Xinjun Lu

Academic Editor

PLOS ONE

Journal Requirements:

Reviewers' comments:

Reviewer's Responses to Questions

**Comments to the Author**

1. Is the manuscript technically sound, and do the data support the conclusions?

Reviewer #1: Partly

Reviewer #2: Yes

Reviewer #3: Yes

2. Has the statistical analysis been performed appropriately and rigorously? 

Reviewer #1: Yes

Reviewer #2: Yes

Reviewer #3: Yes

3. Have the authors made all data underlying the findings in their manuscript fully available?

Reviewer #1: Yes

Reviewer #2: Yes

Reviewer #3: Yes

4. Is the manuscript presented in an intelligible fashion and written in standard English?

Reviewer #1: Yes

Reviewer #2: Yes

Reviewer #3: Yes

5. Review Comments to the Author

Reviewer #1: In this study, the Authors aimed to identify shared molecular mechanisms and potential drug targets by comparing the gene expression profiles of 6 different types of digestive system cancers including cholangiocarcinoma, colorectal cancer, esophageal carcinoma, hepatocellular carcinoma, pancreatic adenocarcinoma and stomach adenocarcinoma. The Cancer Genome Atlas (TCGA) database was explored to obtain transcriptomic data for 6 types of digestive system cancers. A filtering criterion was applied to identify genes with an expression level of 20 or above in at least 180 samples, yielding a dataset of 17,673 genes.

The combination of 12 cholangiocarcinoma and hepatocellular carcinoma exhibited the highest number of shared DEGs, with 738 common genes, while colorectal cancer (CRC) and LIHC shared 358 DEGs, ranking sixth. The results indicate that the liver-intestinal interaction involves the highest number of small molecules, totaling 1147 and the highest absolute values of small molecule matching scores. These drugs were selected for their potential to target the molecular underpinnings of CRC and LIHC, offering a promising avenue for further research and development. Of interest, the study results revealed the shared molecular pathways that may be targeted by Amonafide and BX795, offering a potential rationale for their use in a combined therapeutic strategy for CRC and LIHC.

The study is of interest exploring novel strategy to identify potential target(s) of anti-cancer agents. However some issues need further information and should be addressed.

-Important data for study analyses were obataind form the TCGA databse. However, considering the patway of LIHC, it is well known that the etiology of underlying liver disease (viral, non-viral, meabolic, autoimmune chronic liver disease) may affect the cancer development and patway. Therefore, if available, it would be of major relevance to report the etiology of enrolled LIHC patients.

-Of interest, the authors found that the core proteins of NXF1 and NCBP2 in the Ubiquitin Mediated Proteolysis pathway, and RPS27A as a core protein in RNA Translation pathway, were also drive genes of CRC and LIHC and these pathways may hold the potential for targeted therapeutic intervention that leverages the shared molecular mechanisms of the two cancers. Importantly, capecitabine, a prodrug of 5-FU has anti-tumor efficacy not only in CRC but also has demonstrated a good anti-tumor efficacy in hepatocellular carcinoma patients failing sorafenib therapy as recently demonstrated (doi: 10.1007/s00432-017-2556-6.). This may represent a further proof of common therapeutic target(s) and is therefore worth to be recalled and discussed.

Reviewer #2: 1. Were toxicity assays performed on normal epithelial cells, such as FHC for colon or THLE-2 for liver?

2. Could long-term treatment induce resistance (e.g., via ABC transporters)?

3. The authors must State whether replicates were technical or biological (e.g., n = 3 independent experiments

4.Compare drug efficacy to standard therapies (e.g., 5-FU for CRC, sorafenib for LIHC).

5.Figure 1: UMAP/hierarchical clustering suggests homogeneity, but the text later highlights CRC-LIHC differences. Clarify this discrepancy!!!!

Reviewer #3: Dear Researchers,

I have reviewed your manuscript and here are some comments and suggestions:

Title: The title adequately reflects the objectives of the study and is of appropriate length.

Abstract: The abstract is concise and perfectly presents each key aspect of the study, fulfilling its purpose.

Introduction: I suggest a more in-depth discussion of the current need to find therapies for cancers in general, and then focus on gastric cancer. It would be beneficial to include more detail on the impact of current treatments and how drug repositioning could address these issues. In addition, they might consider mentioning the economic benefits of drug repositioning in oncology.

Methodology: The methodology is adequate and well detailed, clearly explaining each experiment and each analysis performed.

Results: I suggest going a little deeper into the results obtained using the Connectivity Map (C-Map) and the impact of the small molecules identified. This would help to provide a greater understanding of how these results could be applied in clinical settings.

Discussion: I propose to further address the clinical impact of the findings of this research. It would be useful to explore how the compounds Amonafide and BX795 could be integrated into combination or alternative treatments. In addition, it would be valuable to discuss the limitations of the in vitro experiments performed and possible directions for future in vivo validations. I also suggest including an analysis of the economic benefits of drug repositioning and its potential impact on the pharmaceutical industry and correlate this with their findings.

Conclusions: I recommend subtly adding a mention of the prospects towards clinical trials, to highlight the relevance of the results obtained and their possible future application in clinical practice.

References: I have no additional comments on the references.

I sincerely congratulate the excellent work presented in your manuscript. The research promises great advances in cancer treatment through drug repositioning. Your effort and dedication are evident, and I encourage you to continue this valuable work.

Sincerely,

SAD

6. PLOS authors have the option to publish the peer review history of their article (what does this mean? ). If published, this will include your full peer review and any attached files.

**Do you want your identity to be public for this peer review?** For information about this choice, including consent withdrawal, please see our Privacy Policy .

Reviewer #1: No

Reviewer #2: **Yes: ** Sakarie Mustafe Hidig MD

Reviewer #3: **Yes: ** Sergio Ayala-Diaz

---

## [Author Response · Author response to Decision Letter 1]

2 May 2025

Letter to the Editor

Subject: Revise of Manuscript for Publication Consideration

Dear Editor,

Enclosure Notification

Thank you for the opportunity to revise our manuscript titled “Drug Repositioning for Pan-Cancers of the Digestive System: Identification of Amonafide and BX795 as Potential Therapeutics via Integrative Omics Analysis” in response to the comments and suggestions provided by the reviewer. We appreciate their valuable feedback and have made significant improvements to address their concerns.

We have carefully considered each comment raised and have made the following revisions to the manuscript:

Response to Reviewer 1:

- Question 1: Important data for study analyses were obataind form the TCGA databse. However, considering the patway of LIHC, it is well known that the etiology of underlying liver disease (viral, non-viral, meabolic, autoimmune chronic liver disease) may affect the cancer development and patway. Therefore, if available, it would be of major relevance to report the etiology of enrolled LIHC patients.

- Revision: This is a very good suggestion. This is the deficiency of this article, which lacks a more detailed analysis of liver disease types. Moreover, the clinical information for specific patients in TCGA is incomplete and therefore cannot be provided.

- Question 2: Of interest, the authors found that the core proteins of NXF1 and NCBP2 in the Ubiquitin Mediated Proteolysis pathway, and RPS27A as a core protein in RNA Translation pathway, were also drive genes of CRC and LIHC and these pathways may hold the potential for targeted therapeutic intervention that leverages the shared molecular mechanisms of the two cancers. Importantly, capecitabine, a prodrug of 5-FU has anti-tumor efficacy not only in CRC but also has demonstrated a good anti-tumor efficacy in hepatocellular carcinoma patients failing sorafenib therapy as recently demonstrated (doi: 10.1007/s00432-017-2556-6.). This may represent a further proof of common therapeutic target(s) and is therefore worth to be recalled and discussed.

- Revision: It has been added in lines 53-55.

Response to Reviewer 2:

- Question 1: Were toxicity assays performed on normal epithelial cells, such as FHC for colon or THLE-2 for liver?

- Revision: This article did not conduct toxicity assays. The candidate drugs were approved for marketing by the FDA, and experiments on the corresponding cells were also mentioned in the references. We will pay more attention to this aspect in the subsequent research.

- Question 2: Could long-term treatment induce resistance (e.g., via ABC transporters)?

- Revision: This is a very good suggestion. However, this article aims to discover new drugs through drug reuse. In subsequent studies, we will pay more attention to this aspect.

- Question 3: The authors must State whether replicates were technical or biological (e.g., n = 3 independent experiments

- Revision: Biological repetition, mentioned in the Method section 185; Line 194, etc.

- Question 4: Compare drug efficacy to standard therapies (e.g., 5-FU for CRC, sorafenib for LIHC).

- Revision: This is a very good idea. Due to the large number of cell experiments in this article and time constraints, it is impossible to supplement. This limitation has been pointed out in the subsequent discussion section.

Line 440�Moreover, our research lacks Compare drug efficacy to standard therapies, but these drugs do have inhibitory effects on two types of cancers.

- Question 5: Figure 1: UMAP/hierarchical clustering suggests homogeneity, but the text later highlights CRC-LIHC differences. Clarify this discrepancy!

- Revision: Because UMAP/ hierarchical clustering was analyzed based on the whole transcriptome, it showed that different cancer tissues were consistent as a whole. However, in the subsequent differential genes and gene expression modules, it was demonstrated that there were differences in CRC-LIHC, and common regulatory modules were found.

Response to Reviewer 3:

- Question 1: Introduction: I suggest a more in-depth discussion of the current need to find therapies for cancers in general, and then focus on gastric cancer. It would be beneficial to include more detail on the impact of current treatments and how drug repositioning could address these issues. In addition, they might consider mentioning the economic benefits of drug repositioning in oncology.

- Revision: It has been added in lines 45-61.

- Question 2: Results: I suggest going a little deeper into the results obtained using the Connectivity Map (C-Map) and the impact of the small molecules identified. This would help to provide a greater understanding of how these results could be applied in clinical settings.

- Revision: All the matched obtained small molecules and their target genes are mentioned in Supplementary table S8, and some literature on the role of small molecules in cancer research is provided.

- Question 3: Discussion: I propose to further address the clinical impact of the findings of this research. It would be useful to explore how the compounds Amonafide and BX795 could be integrated into combination or alternative treatments. In addition, it would be valuable to discuss the limitations of the in vitro experiments performed and possible directions for future in vivo validations. I also suggest including an analysis of the economic benefits of drug repositioning and its potential impact on the pharmaceutical industry and correlate this with their findings.

- Revision: It has been added in lines 375-391.

- Question 4: Conclusions: I recommend subtly adding a mention of the prospects towards clinical trials, to highlight the relevance of the results obtained and their possible future application in clinical practice.

- Revision: It has been modified on line 453-455. These findings provide a foundation for future research and these compounds are known clinical research drugs. Drug repurposing may change the treatment prospects for patients with these diseases.

Thank you for your time and consideration. We look forward to your positive response.

Sincerely,

Fei Gao, Ph.D.

Principal Investigator

HIM-BGI Omics Center

Zhejiang Cancer Hospital

Hangzhou Institute of Medicine (HIM)

Chinese Academy of Sciences (CAS)

Hangzhou, China

Email: [flys828@gmail.com]

---

## [Decision Letter · Decision Letter 1]

18 May 2025

Drug Repositioning for Pan-Cancers of the Digestive System: Identification of Amonafide and BX795 as Potential Therapeutics via Integrative Omics Analysis

PONE-D-25-07116R1

Dear Dr. Gao,

We’re pleased to inform you that your manuscript has been judged scientifically suitable for publication and will be formally accepted for publication once it meets all outstanding technical requirements.

Kind regards,

Xinjun Lu

Academic Editor

PLOS ONE

Reviewers' comments:

Reviewer's Responses to Questions

**Comments to the Author**

1. If the authors have adequately addressed your comments raised in a previous round of review and you feel that this manuscript is now acceptable for publication, you may indicate that here to bypass the “Comments to the Author” section, enter your conflict of interest statement in the “Confidential to Editor” section, and submit your "Accept" recommendation.

Reviewer #1: All comments have been addressed

Reviewer #2: All comments have been addressed

2. Is the manuscript technically sound, and do the data support the conclusions?

Reviewer #1: Yes

Reviewer #2: Yes

3. Has the statistical analysis been performed appropriately and rigorously? 

Reviewer #1: Yes

Reviewer #2: Yes

4. Have the authors made all data underlying the findings in their manuscript fully available?

Reviewer #1: Yes

Reviewer #2: Yes

5. Is the manuscript presented in an intelligible fashion and written in standard English?

Reviewer #1: Yes

Reviewer #2: Yes

6. Review Comments to the Author

Reviewer #1: The authors have provided in the revised manuscript the necessary changes and the manuscript can be now accepted

Reviewer #2: The authors have provided clear and satisfactory responses to all of my questions. I find the article acceptable for publication.

7. PLOS authors have the option to publish the peer review history of their article (what does this mean? ). If published, this will include your full peer review and any attached files.

**Do you want your identity to be public for this peer review?** For information about this choice, including consent withdrawal, please see our Privacy Policy .

Reviewer #1: No

Reviewer #2: **Yes: ** Sakarie Mustafe Hidig MD

---

## [Editor Report · Acceptance letter]

PONE-D-25-07116R1

PLOS ONE

Dear Dr. Gao,

I'm pleased to inform you that your manuscript has been deemed suitable for publication in PLOS ONE. Congratulations! Your manuscript is now being handed over to our production team.

Kind regards,

on behalf of

Dr. Xinjun Lu

Academic Editor

PLOS ONE